# Underwater Noise Assessment in the Romanian Black Sea Waters

**Maria Emanuela Mihailov** [1,*]🄳, **Gianina Chirosca** [2,3]🄳 **and Alecsandru Vladimir Chirosca** [2,*]🄳

1 Maritime Hydrographic Directorate "Comandor Alexandru Catuneanu", Fulgerului Street no. 1, 900218 Constanta, Romania

2 Faculty of Physics, University of Bucharest, Atomiștilor 405, 077125 Magurele, Romania; gianina.chirosca@inoe.ro

3 National R&D Institute for Optoelectronics "INOE 2000", Atomistilor 409, 077125 Magurele, Romania

* Correspondence: emanuela.mihailov@dhmfn.ro (M.E.M.); alecsandru.chirosca@unibuc.ro (A.V.C.)

**Abstract:** The Black Sea, a unique semi-enclosed marine ecosystem, is the eastern maritime boundary of the European Union and holds significant ecological importance. The present study investigates anthropogenic noise pollution in the context of the Marine Strategy Framework Directive's Descriptor 11, with a particular emphasis on the criteria for impulsive sound (D11C1) and continuous low-frequency sound (D11C2) in Romanian ports, which handle a substantial share of regional cargo traffic, and impact maritime activities and associated noise levels. The noise levels from shipping activity vary across Romanian waters, including territorial waters, the contiguous zone, and the Exclusive Economic Zone. These areas are classified by high, medium, and low ship traffic density. Ambient noise levels at frequencies of 63 Hz and 125 Hz, dominated by shipping noise, were established, along with their hydrospatial distribution for the 2019–2020 period. Furthermore, predictive modeling techniques are used in this study to assess underwater noise pollution from human sources. This modeling effort represents the first initiative in the region and utilizes the BELLHOP ray-tracing method for underwater acoustic channel modeling in shallow-water environments. The model incorporates realistic bathymetry, oceanography, and geology features for environmental input, allowing for improved prediction of acoustic variability due to time-varying sea variations in shallow waters. The study's findings have important implications for understanding and mitigating anthropogenic noise pollution's impact on the Black Sea marine ecosystem.

**Keywords:** underwater noise; Marine Strategy Framework Directive; Descriptor 11; hydrospatial analysis; Black Sea




## 1. Introduction

Increased levels of anthropogenic noise in the marine environment, broadly attributed to intermittent sources such as shipping, have been shown to negatively impact the ambient sounds, with detrimental consequences for marine fauna and overall ecosystem health [1–3]. Consequently, at low frequencies (from 5 to 500 Hz), maritime traffic, particularly commercial vessels, generates the most significant contribution to the total noise budget [4,5]. Hence, efforts were made to develop quieting technologies for commercial ship vessels [6,7]. The leading causes of radiating hydroacoustic noise are primarily the propeller action, propulsion machinery and hydraulic flow over the hull [8].

The navy has developed and managed advanced underwater acoustic propagation models and acoustic models that focus on acoustic reverberation, acoustic inversion, and target scattering. These models support various applications, such as anti-submarine warfare [9–12] and the localization of underwater vehicles using repeated transmissions of acoustic signals [13–18].

Acoustic numerical models were continuously developed and refined for various ocean/sea basins [19–25] for a given scenario, mainly for environmental impact assessments [26] and their impact on marine mammals [27]. Modeling tools are frequently used

to assess offshore wind farms' noise levels using three-dimensional underwater acoustic propagation models [28]. However, in order for long-range acoustic waveforms using ray-Born modeling [29] to be successfully used in global seismology and explorations, they require the computation of numerous rays from the receiver to the scattering plots.

There has been considerable work conducted on developing standardized monitoring programs for noise levels in European seas [30]. These programs, developed under the Marine Strategy Framework Directive (MSFD), were established on ambient noise indicators centered at 63 Hz and 125 Hz (the frequency bands most likely to be dominated by shipping noise) as part of developing indicators for use within the Marine Strategy Framework Directive (MSFD) [31–33]. The MSFD and its methodological standards aim to achieve Good Environmental Status (GES) by 2020/2026 and establish an ecosystem-based approach with 11 qualitative descriptors for comprehensive marine environmental management [31,33–39]. Within the Marine Strategy Framework Directive (MSFD), Descriptor 11 focuses on mitigating the adverse effects of energy inputs on the marine environment, and GES is achieved when these inputs do not cause adverse changes to the ecosystem [40]. Therefore, Decision 2017/848/E.U. [35] establishes two criteria within Descriptor 11 to assess anthropogenic sound in the marine environment: D11C1 for impulsive sound sources and D11C2 for continuous low-frequency sound levels. The Technical Group on Underwater Noise (TG Noise) [30] plays a key role in supporting member states in implementing Descriptor 11 through the development of standardized monitoring guidance. It highlighted the need for modeling to obtain a complete picture of sound distribution, as monitoring by direct measurements in the marine environment is unavailable for many regions [30].

Acoustic modeling techniques that assimilate automatic ship-tracking (Automatic Identification System–AIS) data have been employed in various projects to generate noise maps for assessing and predicting underwater noise pollution. These techniques utilize automated tracking system data to determine vessel position and estimate their noise emissions, contributing to a better understanding of noise pollution in specific areas [41,42]. The application of AIS-based noise mapping has been under development in joint registers in European projects, such as those for the Mediterranean Sea region: Projects quietMED project—a joint program on underwater noise (D11) for the implementation of the Second Cycle of the MSFD in the Mediterranean Sea [43], and quietMEd2—a joint program for GES assessment on D11-noise in the Mediterranean Marine Region [44], and recently including the Black Sea in QUIETSEAS—assisting cooperation for the implementation of the Marine Strategy Framework Directive on underwater noise [45] project). These registers provide a web–GIS platform with three main functionalities to assess impulsive anthropogenic sound in the marine environment. Users can exchange data, visualize spatial patterns on a map, and calculate indicators aligned with both the MSFD (Criterion 1 of Descriptor 11) or following the Convention for the Protection of the Marine Environment of the North-East Atlantic (the OSPAR Convention) Common Indicator for impulsive noise [46,47].

Despite its unique geographical position as a link between Europe and Asia via the Bosporus and Dardanelles Straits, acoustic pollution does not bypass the Black Sea due to its significant role in global maritime trade. Furthermore, anthropogenic pressures on marine ecosystems, including the increasing extraction of marine resources and the persistence of unsustainable practices, compromise these environments' ecological integrity and resilience.

Consequently, conventional hydrocarbon extraction has become challenging and interest in the exploration and exploitation of oil and gas resources in the Black Sea basin has been growing. However, there are no considerable reserves of hydrocarbon production in shallow waters in Turkey, Romania, and Bulgaria [48]. Impulsive sounds generated by these activities within the low- and medium-frequency (D11C1) in the North-Western Black Sea (NWBS) are primarily generated by the operation of air gun arrays used in seismic exploration for hydrocarbon resources [49]. Still, no detailed quantitative assessment of the Black Sea basin was achieved.

This paper presents the authors' integrated regional study combined with the first modeling achievement applying the BELLHOP ray-tracing method [50] for underwater acoustic channel modeling in shallow water environments. The main aim is to discuss and analyze the geophysical characteristics and underwater noise recordings collected during hydrographic and oceanographic surveys within the NWBS shelf. This analysis will contribute to the assessment of Marine Strategy Framework Directive (MSFD) Descriptor 11, specifically addressing anthropogenic continuous low-frequency sound in water (D11C2), as reported to Marine Reporting Units (MRU).

## 2. Materials and Methods

### 2.1. Study Area

The Black Sea is peculiarly vulnerable to anthropogenic impacts as a significant semi-closed deep internal basin. With significant economic importance for the countries around its coasts, the Black Sea basin supports a booming tourism industry, fisheries, and transport, served by large ports. In addition, many resorts and industrial sites have developed along its coasts. Moreover, oil deposits were discovered, and interest in oil drilling increased [51]. Oil is piped from inland and coastal fields to the Black Sea harbor for transportation, endangering it by oil pollution [52].

The NWBS shelf comprises the entire Romanian offshore sectors and the western part of the Odesa Gulf from the Ukrainian offshore and adjacent onshore regions. According to the Anemone Project Deliverable 1.3 report [53], for the Romanian BS waters, four Marine Reporting Units (MRU) were identified (Table 1, Figure 1).

**Table 1.** MRU for the NWBS shelf (Romanian waters), as MSFD requirements [53].

| Marine Area | Marine Reporting Units | Depth Limits (m) | Area (km$^2$) |
|---|---|---|---|
| Variable salinity | BLK_RO_RG_TT03 [53] | 0–30 | 1358.95 |
| Coastal | BLK_RO_RG_CT [53] | 0–30 | 1040.17 |
| Shelf | BLK_RO_RG_MT01 [53] | 30–200 | 20,164.89 |
| Open sea | BLK_RO_RG_MT02 [53] | >200 | 7058.25 |

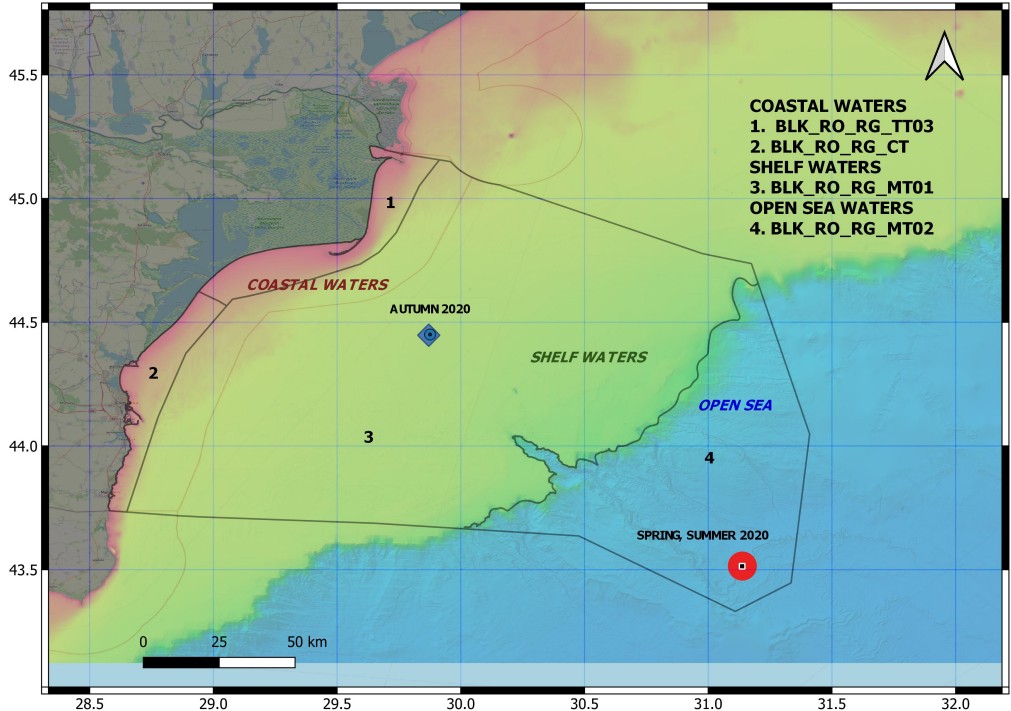

**Figure 1.** North-Western Black Sea (NWBS) shelf bathymetry, Marine Reporting Units (MRU) and CTD stations (spring, summer and autumn).

## 2.2. Sediment Type Characteristics

In terms of crustal structure, the Black Sea basin consists of Western and Eastern rift-type sedimentary basins [54,55]. Both sub-basins are different concerning the time of opening, structure, stratigraphy, and the thickness of their sedimentary fill. The entire basin bottom is subdivided into a shelf—with the sediment pattern governed by surface and longshore bottom currents and wave action [56], continental slope (basin apron), and abyssal (Euxine) plain. On the NWBS shelf area, the Danube sediment supply dispersal pattern indicates two main areas with different depositional processes [57]: supplying the internal shelf and depriving the outer shelf. As highlighted in [57], a very high sedimentological diversity resulted from the grain size analyses, ranging from pure sand to pure clay. The sandy sediments (sand, silty, and clayey sand) appear as a narrow littoral band and as isolated bodies, especially in the *Phyllophora* field [58]. However, most of the NWBS region is dominated by finer sediments, especially silty clays and clayey silts.

The Danube Prodelta area sediment is homogenous, with an upper layer thickness of about 0–15 cm, dominating muds with rare interlayers of silts [57]. In the corresponding area of the Danube influence, a fluid overlays a 3.5–4.5 cm thick sediment enriched in organic matter, or a semiliquid soft layer, represented by a thin layer (1–4 cm) of coccoliths ooze accumulated during the first appearance of *Emiliania huxleyi* in the Black Sea. The grain size composition is relatively homogeneous, dominated by silty clays alternating with clays [57,59]. Shcherbakov et al. (1979) [60] characterized the continental shelf with the following sedimentary facies that can be recognized: (a) *Modiolus* mud, occupying the top of the sedimentary sequence between 50 and 125 m of water depth, is a light-colored mud, very rich in *Modiolus phaseolinus coquinas*, the thickness of which does not usually exceed 30 cm; (b) *Mytilus* mud (*Mytilus galloprovincialis*) from the shelf break down to 50 m water depth; (c) *Dreissena* mud: the surficial sediment formed from shells of *Dreissena*, located from 130 m of water depth. The abrupt transition between *Dreissena* mud and *Mytilus* mud corresponds to the change from fresh/brackish to marine conditions in the Black Sea [61].

The composition and characteristics of seabed sediments play a crucial role in underwater acoustics, influencing sound propagation, attenuation, and scattering. The sediment type, whether it is sand, silt, clay, or a mixture, affects how sound waves interact with the seabed. Different sediments have varying sound absorption and reflection properties, which can significantly impact the transmission loss (TL) and range of underwater sound signals.

In this study, the diverse sediment types in the NWBS shelf, ranging from sands to clays, are considered in order to better understand the sound propagation patterns in the region. The sediment characteristics described inform the selection of appropriate parameters for the BELLHOP model, which is used to simulate sound propagation. To ensure the model's accuracy, input data about the seabed's geo-acoustic properties, including sediment type and sound attenuation coefficients, are required. Furthermore, the distribution of sediment types is related to other environmental factors, such as water depth, currents, and the presence of marine life. Understanding these relationships provides a more comprehensive picture of the underwater environment and its potential impact on sound propagation.

## 2.3. In Situ Conductivity–Temperature–Depth Data

To investigate the interrelationships between salinity, temperature, and sound velocity within the water column of the North-Western Black Sea (NWBS), Conductivity–Temperature–Depth (CTD) profiles were acquired using a Castaway CTD instrument. This dataset facilitated the analysis of seasonal variations in these key physical parameters (Figure 1). Chosen CTD profiles were selected to provide accurate water stratification during the season, at 43.54° N and 31.14° E offshore for the spring and summer seasons and 44.48° N and 29.86° E for autumn (Figure 2). Seawater parameters, including pressure, temperature, and salinity, were measured in situ using the CTD software. Sound speed

was subsequently calculated using the computed CTD data using the Chen and Millero equation [49].

Hydrographic data, including CTD profiles, were collected in 2020 during periodical surveys on the NWBS shelf performed by the MHD onboard R/V "Comandor Alexandru Catuneanu". Golden Software's Surfer software was utilized to generate graphical representations of the datasets, enabling visualization and analysis of spatial variability in the observed parameters.

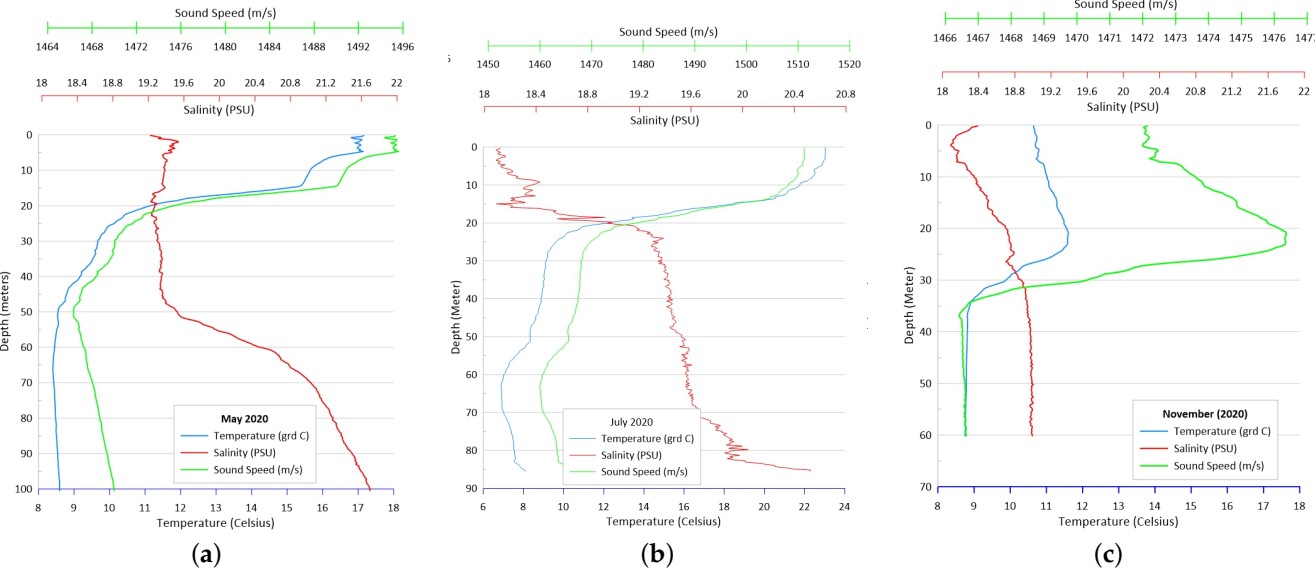

**Figure 2.** Sea temperature, salinity, and sound speed profiles during the 2020 year in (**a**) spring, (**b**) summer, and (**c**) autumn.

## 2.4. Underwater Noise Surveys

Using the autonomous hydrophone system Cetacean Research™'s C55 series [62,63], the deployed passive acoustic generated a considerable amount of raw data containing the required input for the study (ship noise characteristics and environmental sounds such as wind, rain, waves, or mining activity around the study site). Therefore, prior to data analysis, the processing operations were performed to extract the necessary information using SpectraPLUS Spectral Analysis Software 5.0 [64] and PAMGuide using Matlab R2024a environment [65,66].

Multiple processing operations were conducted utilizing SpectraPLUS Spectral Analysis Software [67] to analyze the underwater noise recordings. These operations involved generating spectrograms to visualize the frequency spectrum over time, calculating sound pressure levels (SPLs) to quantify the noise magnitude in decibels (dB) and performing one-third octave band analysis to examine noise levels within specific frequency ranges. Statistical analysis was additionally employed to calculate descriptive statistics such as the mode, 95th percentile, and root-mean-square (RMS) levels, thus providing a comprehensive characterization of the noise data. Sound Exposure Level (SEL) analysis was performed employing PAMGuide [65,66] in the MATLAB environment, a specialized software package for passive acoustic monitoring data analysis that provides tools for calculating SEL metrics.

The system was positioned along the Romanian Black Sea shelf, 2 m above the bottom of the sea (Figure 3). The data comprise one-third octave noise levels with duty-cycled recordings of 10 min on, 10 min off, and ambient noise up to 48 kHz (sampling rate of 96 kHz). Acoustic data were recorded with times varying from site to site, depending on meteorological factors, from 6h deployments in the northern part (for the transitional marine water BLK_RO_RG_TT03) to 24 h in the southern region (for shelf BLK_RO_RG_CT, shelf BLK_RO_RG_MT01 and open sea BLK_RO_RG_MT02 waters). The hydrophone with a transducer sensitivity of −199 dB, re 1 V/μPa, equipped with a protective cage, was deployed only in the summer season.

Sound Exposure Levels (SEL), a metric used to characterize the potential effects of sound on marine mammals, were calculated for 14 underwater acoustic datasets collected at distinct locations (Figure 3). The temporal resolution for the analysis was 1 h, as the data were processed and analyzed on an hourly basis. This temporal resolution was selected to capture the diurnal variability of noise levels. Hourly SEL values were computed for one-third octave bands, encompassing ambient soundscapes and anthropogenic noise sources such as commercial shipping and recreational boating. The distribution of noise levels across the 63 Hz and 125 Hz bands, relevant to MSFD monitoring, exhibited similar ranges.

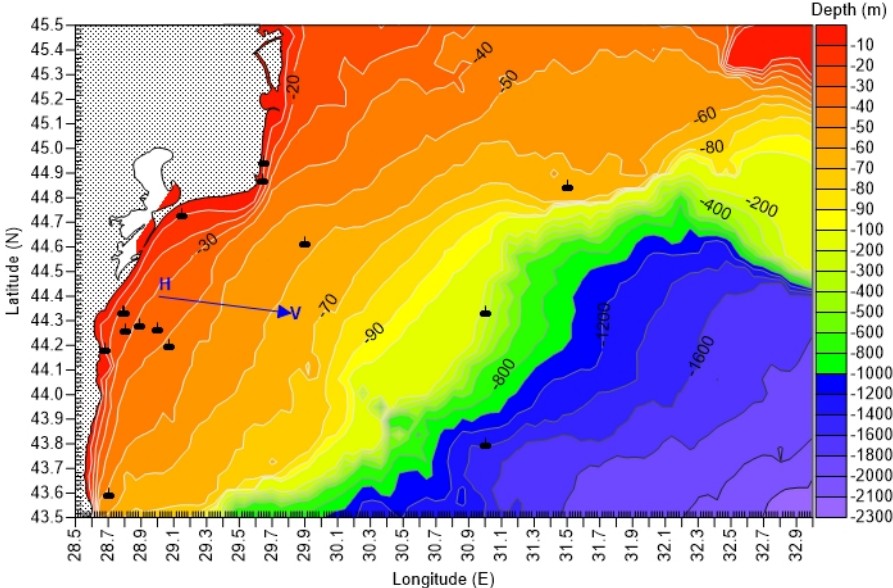

**Figure 3.** Bathymetry for the NWBS shelf and the in situ recording sites (black buoy sign), and the blue arrow line represents the sound propagation transect from the source (H) to the destination (V).

## 2.5. Regional Underwater Noise Modeling

We present the first efforts to predict and exploit the spatial (on-site and local) variability of underwater sound propagation, using BELLHOP, a beam/ray-tracing model [50], for modeling and prediction of the acoustic pressure fields, considering a specific underwater environment that incorporates realistic bathymetry, oceanography, and geology features for the environmental input. The bathymetric input in the model and the section profile combine direct measurement methods and available online databases: EMoDNET Bathymetry [68], GEBCO 2021 [69], and Maritime Hydrographic Directorate (MHD) multi-beam high-resolution data (Figure 3). Merging the available datasets allows us to improve the prediction of acoustic variability due to time-varying sea variations in shallow waters.

A key consideration in underwater acoustic modeling is the choice between coherent and incoherent transmission loss (TL) predictions. Coherent predictions account for phase differences in multipath propagation and are essential for accurately simulating acoustic fields in certain scenarios. Conversely, incoherent predictions, which neglect phase interactions, may be suitable for other applications. The BELLHOP model [50] was selected for this study due to its capability to generate both coherent and incoherent TL estimates, providing flexibility in capturing the complexities of sound propagation in the study area.

The Phyton Programming Language [70] was selected as the environment for the acoustic propagation modeling. The modeling location is established at 30 m water depth at 44.4° N and 29.0° E, which corresponds to the nearest location of the decommissioned Gloria's drilling platform (44.52° N, 29.57° E) at the 50 m bathymetric line [71]. The physical geometry of the sound source is modeled in 601 points as generated by eigenrays that connect the source position with the receiver position in a dependent environment. The geo-acoustic properties of the seabed as the input data in the model were given: the sound speed profile for the warm season (July 2020) and the attenuation coefficient $\alpha$ of

0.5 dB/wavelength (characterizing the sandy sediments in the study area as described in Section 2.2) based on the seafloor sediment database. Also, the effects of bathymetric range-dependence and environmental variability are considered using the section bathymetric profile (Figure 4) and the sea-state (wave height established at 0.5 m).

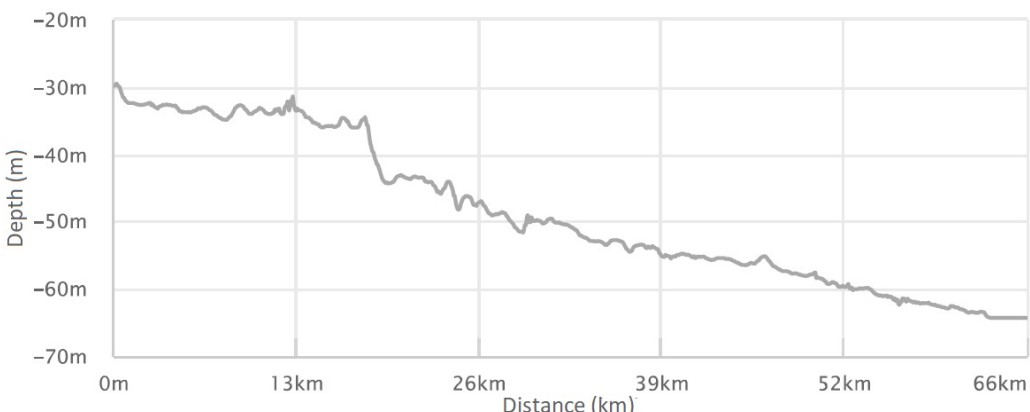

**Figure 4.** The bathymetric profile of the analyzed area, from the coastline on the direction between hydrophone (44.4018° N/29.01449° E) and the vessel (44.3375° N/29.8115° E).

## 3. Results

### 3.1. Seasonal Variability of the Main Physical Parameters

The seasonal variability of seawater temperature and salinity determines the changes in sound velocity and thus affects marine acoustic propagation. The seasonal variability is distinguished in the sound propagation pattern in the NWBS shelf, which is discussed in this section. The characteristic of the sound speed profiles for three seasons, at 43.54° N and 31.14° E offshore for spring and summer seasons and 44.48° N and 29.86° E for autumn, shows that the shallow sound channel occurs in the main thermocline (Figure 2).

Figure 2 shows that the upper boundary of the acoustic layer is found below 15 m (for spring and summer) or 20 m (for early autumn), and the seawater temperature (ST) is mainly affected by sunlight and presents a relatively stable negative gradient. Conversely, the seawater temperature is constant below 50 m or below (depending on the season), called the deep-water isothermal layer. Below the thermocline, the Black Sea is characterized by the Cold Intermediate Layer (CIL), where the sea temperature is constant (isothermal layer) and is strongly dependent on the increase in static pressure [49], presenting an almost linear positive gradient distribution (about 0.03 m/s/m in the warm season, Figure 2b).

### 3.2. Underwater Noise Analysis and Simulation

This section presents the results of the underwater noise assessment conducted in the Western Black Sea and the total transmission loss (TL) simulations obtained using the BELLHOP model. The analysis of in situ data focuses on characterizing the underwater acoustic environment and identifying the main sources and patterns of noise pollution. The findings will contribute to a better understanding of the impact of anthropogenic activities on the marine ecosystem and inform the development of effective management strategies.

In Table 2, descriptive statistics were achieved according to MRU-MSFD regions (Table 1) for noise data recorded on shallow and deep Romanian Black Sea waters. The configuration of the analysis is focused on third-octave band sound pressure levels.

Mode noise levels from the field recordings measurements from 2019 to 2020 on the NWBS shelf (corresponding to the Romanian Black Sea waters) ranged from 67.0 to 74.7 dB re 1 μPa, and the root-mean-square (RMS) level was lower than the 95th percentile. The lower frequency bands are more affected by the very shallow waters in BLK_RO_RG_TT03 (direct Danube influence) and BLK_RO_RG_CT (Table 2).

**Table 2.** Descriptive statistics for recorded underwater noise for MRU Western Black Sea.

| MRU Name | Frequency | Mode (dB) | 95th (dB) | RMS (dB) | SEL (dB) |
|---|---|---|---|---|---|
| BLK_RO_RG_TT03 | 63 Hz | 73.2 | 73.7 | 73.0 | 106.6 |
| BLK_RO_RG_TT03 | 125 Hz | 67.0 | 73.5 | 67.2 | 97.6 |
| BLK_RO_RG_CT | 63 Hz | 71.5 | 92.5 | 78.5 | 105.8 |
| BLK_RO_RG_CT | 125 Hz | 67.3 | 82.4 | 74.3 | 101.7 |
| BLK_RO_RG_MT01 | 63 Hz | 72.8 | 74.3 | 73.4 | 103.8 |
| BLK_RO_RG_MT01 | 125 Hz | 73.5 | 74.7 | 73.5 | 105.4 |
| BLK_RO_RG_MT02 | 63 Hz | 74.7 | 74.7 | 74.3 | 104.0 |
| BLK_RO_RG_MT02 | 125 Hz | 73.5 | 74.8 | 74.5 | 104.2 |

Figure 5 presents the calculated Pressure Spectral Densities (PSD), using Matlab software, for an offshore station from the underwater acoustic data collected for MSFD purposes; it reveals a soundscape dominated by persistent low-frequency sound energy below 10 Hz due to environmental factors such as wind and waves. Intermittent bursts of mid-frequency sound (10 Hz–1000 Hz) are superimposed on this low-frequency background, suggesting biological sources like vocalizing marine mammals or anthropogenic activities such as shipping. These mid-frequency events exhibit variability in both intensity and duration, indicating a range of potential sources or fluctuations in source behavior. High-frequency sounds exceeding 1000 Hz, from events like echolocation clicks, are less frequent and typically shorter in duration. A potential trend towards increased mid-frequency activity over time warrants further investigation to determine if it reflects natural diurnal or seasonal patterns or an increase in anthropogenic noise pollution.

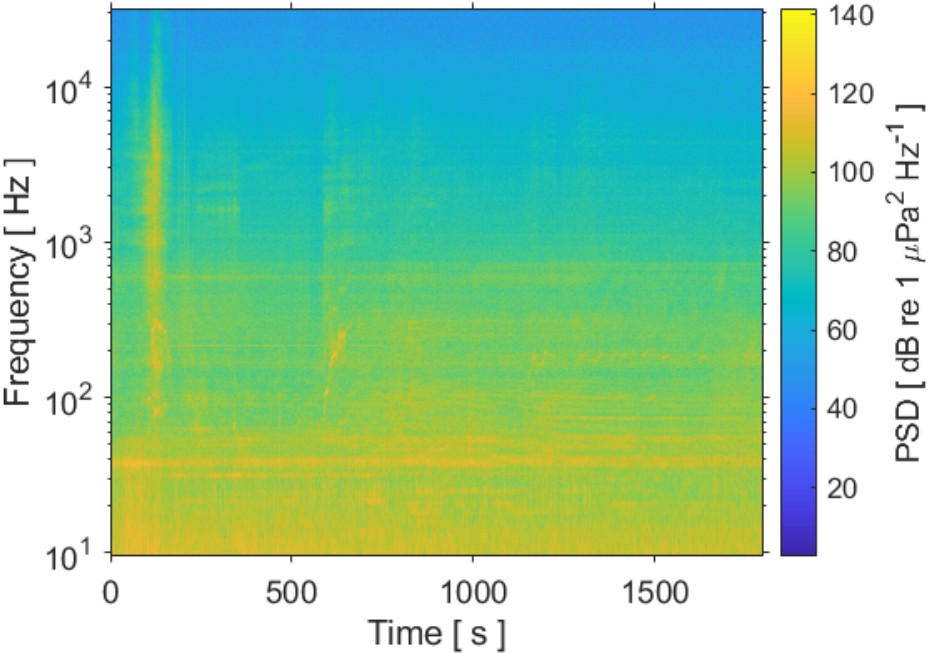

**Figure 5.** Sample of a spectrogram of recorded data using Cetacean Research Hydrophone system, evidencing the presence of marine mammal activity in a mixed ambient and anthropogenic soundscape in the offshore NWBS.

The results shown in Figure 6 are the total transmission loss (TL) simulations for phase-coherent and acoustic transmission obtained using the BELLHOP model, with a source positioned at 0 m depth and 30 m depth. It highlights that the waveguide formed from a negative to positive sound speed profile around the 15 m depth, propagating with a range through more than 200 m. The modeling results significantly contribute to the

future underwater noise hydrospatial assessments in the NWBS and provide a basis for seeking future trends in the interest area. Our computational environment considers 1000 m between the source and the destination for different frequencies 10 Hz, 63 Hz, 125 Hz, and 1 kHz.

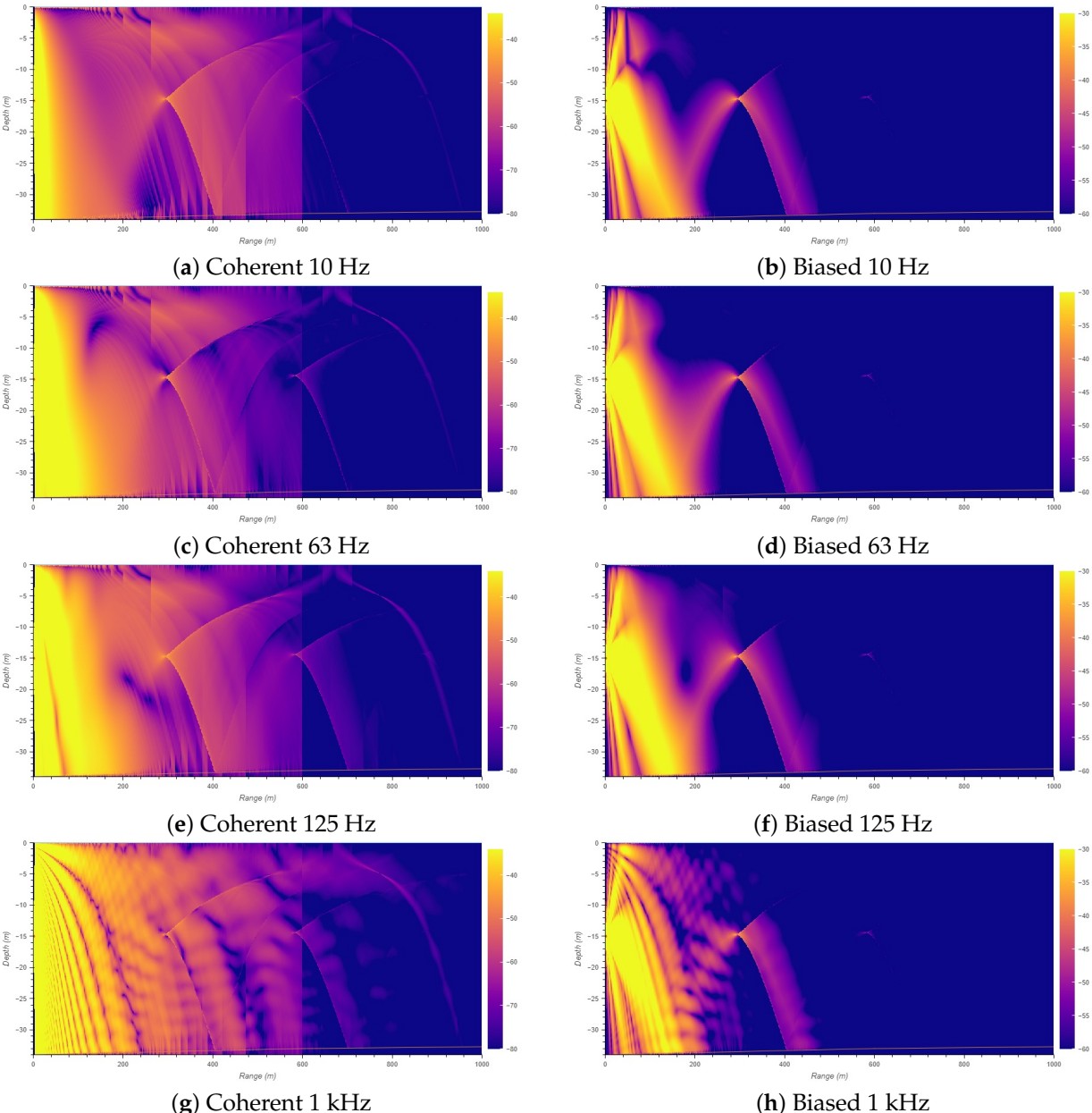

**Figure 6.** Output computation of the transmission loss for coherent and biased waves for a range distance up to 1000 m for 10 Hz, 20 Hz, 63 Hz, 125 Hz, 1 kHz.

The range of 1 km is limited for computing due to the processor performances, as the execution time is considerably high. Moreover, while the computation is performed for one omnidirectional source with a constant intensity distribution within all directions for coherent (taking into account wave interference) and incoherent (not taking into account wave interference), the propagation loss mode tends to be less accurate in the results.

Figure 7 illustrates the sound intensity distribution for a biased source employed in the transmission loss modeling and the directivity pattern indicating the relative sound intensity emitted in different directions. The radial axis of the plot represents the sound intensity level in decibels (dB). The angular axis, in degrees, indicates the direction of

sound propagation relative to a reference point, typically aligned with the direction of maximum intensity. The irregular shape suggests that the biased sound source exhibits a non-uniform sound radiation pattern, indicating that the sound intensity varies depending on the direction of propagation. This directivity pattern is characteristic of more complex sound sources, where the sound radiation is influenced by source geometry and the nearby environment.

Figure 8 presents a visualization of eigenrays generated by the BELLHOP model for a scenario with a sound source at a depth of 15 m and a receiver at a depth of 10 m, where the orange dot marks the location of the sound source, while the blue dot marks the receiver position. The x-axis represents the horizontal range in meters, while the y-axis represents the depth in meters. The blue line depicts the water surface, while the brown line at the bottom represents the seabed. The gray lines illustrate the paths of sound rays as they propagate through the water column, undergoing reflections at both the surface and the seabed. The curvature of these rays indicates the influence of the sound speed profile on sound propagation. The figure highlights the complex interactions of sound waves in a shallow water environment, evidencing phenomena such as refraction and reflection. The varying density of eigenrays in different regions suggests areas of varying sound intensity. This visualization aids in understanding the intricacies of sound propagation and contributes to assessing the underwater noise levels in the NWBS.

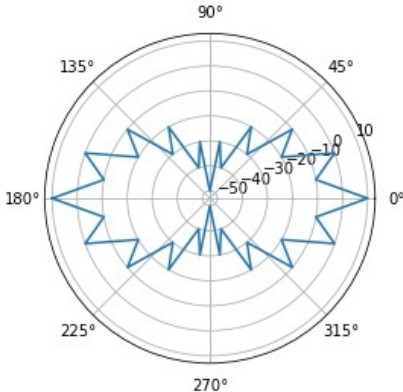

**Figure 7.** Sound intensity distribution for the biased source used for modeling transmission loss.

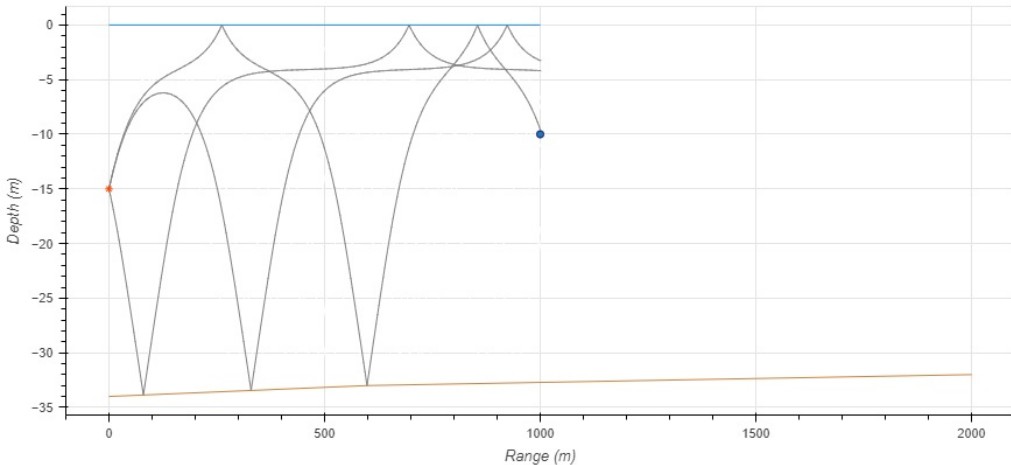

**Figure 8.** Illustration of the eigenrays between the transmitter and receiver in NWBS shallow waters. The orange dot marks the location of the sound source (emitter), while the blue dot marks the receiver position.

## 4. Discussion

Romania, as a European Union member state, should develop activities to achieve the "Good Environmental Status" (GES) of its marine waters following the Commission Decision 2010/477/E.U. [33]. The MSFD outlines a framework for community action in the field of marine environmental policy, including Descriptor 11, which specifically addresses underwater noise pollution.

Despite the MSFD's implementation in 2010, a 2012 assessment revealed insufficient data for meeting the D11 criteria in Romanian Black Sea waters. This lack of data reflects underwater noise pollution's complexity and multi-dimensional nature, exacerbated by human activities and the region's unique bathymetric characteristics (Figure 3). Monitoring and assessing underwater noise pollution in this region presents significant challenges due to factors such as high turbidity and heavy marine traffic.

In addition to the MSFD [31,32], the Maritime Spatial Planning Directive (MSP) [60] promotes an ecosystem-based approach to marine policy. This approach emphasizes the integration of environmental considerations into marine planning and management. In Romania, the MSP framework has been applied to identify potential conflicts among marine users and map anthropogenic activities, including shipping, oil and gas exploitation, fisheries, and tourism, in relation to the environmental status [72].

The anthropogenic noise assessment is a relatively recent development as a pollutant in the Black Sea basin. The impact on marine fauna in this region, including several marine mammal and fish species, is not well assessed due to scarce data on underwater noise recorded data, and all relevant connections for GES are based only on bibliographical data from the ocean noise literature investigating its impact [73]. In the frame of several projects, such as the Monitoring Study Contract by the Romanian Ministry of Water and Forests no.55/2018, the Nucleus Programme PN16230102 during 2016–2017, and the CeNoBs Project—supporting MSFD implementation in the Black Sea through establishing a regional monitoring system of cetaceans (D1) and noise monitoring (D11) for achieving GES [8,32,73], several underwater noise measurements were performed to fill the lack of background data and to develop national expertise in implementing effective underwater noise monitoring. Our interest region still lacks available information on background noise data to comply with MSFD requirements and modeling tools should be developed for the NWBS shelf in regards to D11C1 and D11C2 (EC 2017) and following the Technical Group on Underwater Noise (TG-Noise) recommendations [30,74,75].

Underwater acoustic transmission loss modeling and prediction are key in generating situational awareness in complex navy operations and assisting specific underwater operations. This paper implemented an unclassified acoustic prediction model (BELLHOP) from the freeware version to provide ray trace, transmission loss, travel time, and arrival angle predictions. This model will provide all the output products necessary to support active and passive applications. The required inputs are sound velocity profiles, bathymetry, geometry, and sediment characteristics, as mentioned in regional underwater noise modeling Section 2.5.

Seasonally, the prevailing conditions for sound propagation (SP) in the Black Sea in the sound acoustic channel can be described as follows: winter—positive refraction; summer—negative refraction and propagation; autumn—a surface channel and propagation [49]. During the cold season, strong winds and specific low air temperatures produce a constant sound speed (isovelocity) condition [49,76], resulting in more omnidirectional propagation. Therefore, in contrast to downward refracting propagation in the summer, there is less bottom reflection loss from smaller incident angles (observed in Figure 6). Furthermore, the water column is characterized by a strong stratification during the warm season. As a result, the thermocline induces significant negative refraction and propagation and generates downward refracting propagation conditions. This negative sound speed gradient causes acoustic rays to bend toward the seafloor and interact with the bottom at large angles creating a higher bottom reflection loss.

As observed in Table 2, lower frequencies demonstrate increased susceptibility to the effects of shallow water environments in certain Marine Reporting Units (MRUs). The specific frequencies under consideration are the 63 Hz and 125 Hz one-third octave bands, which are central to this analysis due to their significance in MSFD monitoring and the predominance of shipping noise within these bands. Shipping noise is identified as the primary source contributing to the energy content in these lower frequencies. The elevated noise levels detected in these frequency bands within the shallow water MRUs (BLK_RO_RG_TT03 and BLK_RO_RG_CT) are likely a consequence of several interconnected factors. The reduced water depth in these environments can lead to a higher concentration of sound energy due to reflections and interactions with the seabed and surface. Furthermore, the proximity of the BLK_RO_RG_TT03 to the Danube River introduces potential additional noise sources and influences sound propagation patterns. The unique coastal and shelf characteristics of both BLK_RO_RG_TT03 and BLK_RO_RG_CT including specific bathymetric features and sediment types, can also influence sound propagation, and contribute to the observed higher noise levels. To enhance the statistical analysis, a comparison of noise levels between the 63 Hz and 125 Hz bands across different MRUs has been incorporated. This comparative analysis serves to highlight the variability of noise levels as a function of depth and location.

Visual comparisons with the Transmission Loss (TL) field are presented at 10 Hz, 20 Hz, 63 Hz, 125 Hz, and 1 kHz frequencies, and it seems clear that the BELLHOP model matches the source location and strength of shadow zones. Therefore, we can assume that the lack of high-quality bottom analysis is the most significant limitation on accuracy in the present model configuration. However, the derived propagation paths show that those near the source frequently interact with the bottom, and the computed TL is sensitive to the bottom loss derived from the very shallow selected profile. In 2D noise propagation, theoretical approaches usually consider a uniform directivity pattern, while most noise sources use a specific pattern for the sound directivity, focused in forward and backward directions with an angle span lower than 50 deg. As a result, the sounds propagate mainly in the backward and forward directions, while only a limited fraction of the sound propagates in the normal direction to this axis. Moreover, the distribution can be uniform (with the same intensities among all angles) or biased (generally, the sound is distributed among specific angles). The directivity pattern associated with a given receiver's configuration refers to their sensitivity, and the directivity pattern associated with the sources indicates the sound sources' strength. While the incoherent models tend to lose details by not considering the interferences between the propagated waves, we will focus on the coherent ones [50]. The coherent comparisons with incoherence showed good agreement in the placement of lobes (spikes) in the energy discharge into the shadow zones as the duct decayed. This comparison allows us to implement the BELLHOP in the low-frequency band for the NWBS to provide all the applicable outputs that may form the basis for both passive and active operations.

We performed the transmission loss for this study's omnidirectional and biased sound sources. A directivity pattern is obtained, and the modeling results are plotted on a polar coordinate. The biased source has spherically uniform acoustic radiation, as shown in Figure 7.

Following the prevailing conditions for SP in our interest area, the summer season (which has strong stratification as is shown in Figure 2) was chosen to model the ray trajectories. The model computes the many rays' trajectories with the starting angles at the source, covering the total water volume in the range and depth of interest to the analysis. Moreover, as can be observed in Figure 8, the eigenrays are grouped in downward arriving (DA), upward arriving (UA) and under line-of-sight (LoS) between transmitter and receiver. Furthermore, microscatterers characterize the roughness of sea surface and sea bottom, as is illustrated in Figures 6 and 8, and are distinctive for the shallow seawater acoustic channel. From our modeling results, it can be concluded that the incoherent model can be used in a situation where there is one path from a source to a receiver, and the coherent

model, which is more similar to channels in the natural shallow water environment, is the best one as it balances accuracy and efficiency the situation with multipath propagation.

## 5. Conclusions

With an area of approximately 413,000 km$^2$, the Black Sea binds Eastern Europe to Western Asia, and six riparian countries surround it: Bulgaria, Romania, Ukraine, Russia, Georgia, and Turkey. As the eastern maritime frontier of the European Union, the Black Sea connects with the Mediterranean Sea via the Bosphorus Strait to the Sea of Marmara and then via the Dardanelles Strait to the Aegean Sea part of the Mediterranean Sea. The global pattern in marine traffic for a period of 6 months (January–June 2020) presents different spatial footprints strongly dependent on maritime sectors: cargo and tankers were widespread along main shipping lines, and fishing and recreational vessels were more dispersed between coastal and offshore waters, while passenger's vessels presented a more limited distribution [77].

This study investigated underwater noise pollution in the Western Black Sea, contributing to the fulfillment of the Marine Strategy Framework Directive (MSFD) Descriptor 11 requirements. By analyzing in situ noise data and employing acoustic modeling techniques, the research provides valuable insights into the spatiotemporal variability of underwater noise and its potential impact on the marine ecosystem. The study's findings highlight the significant contribution of anthropogenic activities, particularly shipping, to underwater noise pollution. The dominance of shipping noise at 63 Hz and 125 Hz increases the need for mitigation strategies targeting these frequencies. The spatial variability of noise levels, influenced by vessel traffic density and proximity to ports, emphasizes the importance of targeted management efforts in high-noise areas.

This paper has used the BELLHOP model to analyze the eigenrays and obtain an improved understanding of the underwater acoustic propagation in the NWBS shallow waters. Implementing the BELLHOP ray-tracing model demonstrates its effectiveness in simulating sound propagation in shallow water environments. The model's ability to incorporate realistic bathymetry, oceanography, and sediment features enhances its accuracy in predicting acoustic variability, contributing to a better understanding of sound propagation patterns in the NWBS. Despite the limited range of model simulations (1 km) due to computational constraints, the study successfully analyzed noise levels and their variability. The findings contribute to a better understanding of underwater noise pollution in the studied area, and the author aims to continue the research, focusing on advancing the range of model simulations and investigating the long-term impact of noise pollution on the marine ecosystem.

As a lesson learned from other regional European seas and projects, it is necessary to continue with further research in the interest area with real-time measurements and modeling noise propagation in the Black Sea at the regional scale.

**Author Contributions:** Conceptualization, M.E.M.; methodology, M.E.M. and A.V.C.; software, M.E.M., A.V.C. and G.C.; validation, M.E.M. and A.V.C.; formal analysis, G.C.; investigation, M.E.M.; resources, M.E.M.; writing—original draft preparation, M.E.M., A.V.C. and G.C.; writing and editing, M.E.M., A.V.C. and G.C.; writing—review and editing final version, M.E.M. All authors have read and agreed to the published version of the manuscript.

**Funding:** This research received no external funding.

**Institutional Review Board Statement:** Not applicable.

**Informed Consent Statement:** Not applicable.

**Data Availability Statement:** The data presented in this study are available on request from the authors.

**Acknowledgments:** In this paper, authors use the initial results presented as a short-oral presentation and e-poster at EMSO Time Series Conference 2021 "Observing Ocean Sound", 20–22 October 2021, Canary Islands, Spain2021 [78]. The presented initial results at EMSO Conference has been carried

out with financial support from the Sectorial Research-Development Plan of the Romanian Ministry of National Defence, PSCD 2021–2024 Project (097/2021, 092/2022, 097/2023): *"Development of an integrated monitoring system to increase the quality of hydro-oceanographic data in the area of responsibility of the Romanian Naval Forces"*. The authors would like to thank the anonymous reviewers and members of the editorial team for their comments and contributions. Thanks are extended to the relevant departments of INOE-2000 for their help through the *Core Program with the National Research Development and Innovation Plan 2022-2027, with the support of MCID, project no. PN 23 05/2023 contract 11N/2023, and Program I – Development of the National R&D System, Subprogram 1.2 Institutional Performance-Projects for Excellence Financing in RDI, contr. 18PFE/2021.*

**Conflicts of Interest:** The authors declare no conflicts of interest.

## Abbreviations

The following abbreviations are used in this manuscript:

| | |
|---|---|
| MDPI | Multidisciplinary Digital Publishing Institute |
| DOAJ | Directory of open access journals |
| TLA | Three letter acronym |
| LD | Linear dichroism |
| MSFD | Marine Strategy Framework Directive |
| GES | Good Environmental Status |
| TGNoise | Technical Group on Underwater Noise |
| AIS | Automatic Identification System |
| quietMED | A Joint program for GES assessment on D11-noise in the Mediterranean Marine Region Project |
| quietMEd2 | A Joint program for GES assessment on D11-noise in the Mediterranean Marine Region Project |
| QUIETSEAS | Assisting cooperation for the implementation of the Marine Strategy Framework Directive on underwater noise Project |
| OSPAR | Convention for the Protection of the Marine Environment of the North-East Atlantic |
| BS | Black Sea |
| NWBS | North-Western Black Sea |
| MRU | Marine Reporting Units |
| TL | Transmission Loss |
| CIL | Cold Intermediate Layer |
| MHD | Maritime Hydrographic Directorate |
| SEL | Sound Exposure Levels |
| UN | Underwater Noise |
| MSP | Maritime Spatial Planning Directive |
| CeNoBs | Support MSFD implementation in the Black Sea through establishing a regional monitoring system of cetaceans (D1) and noise monitoring (D11) for achieving GES Project |
| DA | downward arriving |
| UA | upward arriving |
| LoS | under line-of-sight |
| CTD | Conductivity–Temperature–Depth Instrument |
| ST | Seawater temperature |
| PSD | Pressure Spectral Densities |

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
