# Peer review of "Underwater Noise Assessment in the Romanian Black Sea Waters"

_environments, doi:10.3390/environments11120262_

Round 1
Reviewer 1 Report
Comments and Suggestions for Authors
Line 129: citation should be placed at the end of the entire sentence if it refers to the description of the site.
Line 136 cite the reference as Name et al. [56]…
Section 2.3. Being included in the Methods, rather than describing the sound velocity profiles, this section should describe how they were performed, including software used and the source of oceanographic data.
Section 2.4: this section lacks the description of the passive acoustic monitoring system used and the authors should improve the description of both acoustic data collection and analysis. See some specific comment below:
Line 165: Besides the required review of English grammar in this sentence, the choice of the word “required” is debatable considering that any raw acoustic data collected in natural marine environment would contain a vast number of sounds of different origin.
Line 167: it is not clear to which “processing operations” the authors are referring to.
Line 168: the authors should clarify for how many hours rather than stating for “several”.
Similarly, in lines 171-173, to validate and contextualize described statistics, it would be essential to specify the different recording times for each recording location.
Line 175: The authors should clarify whether the acoustic data presented were collected and analyzed by other groups (e.g., MHD) and thus only the output sound levels were used as input for the model validation, or if the data were provided by MHD and analyzed within the context of this study. If the latter is the case, Table 2 and all results reported here should be moved to the Results section.
Line 193-194 : this sentence should be revised and further discussed with the other results in the Results/Discussion
Line 196: I suggest avoiding first person use within the manuscript text. In addition, such types of statements should be reserved for abstract or discussion/conclusions section.
Line 208: include reference to Gloria’s drilling platform.
Line 221-222 this sentence does not fit in the Results section. The authors should state their goals within the Introduction section and discuss them in relation to the Results within the “Discussion”.
Line 222-226 : this information should be included within the previous section in the Methods (2.5.)
Line 229: 0-30 m seems to indicate that you performed TL simulation over the whole depth range. Please correct with “at 0 m depth and at 30 m depth”.
The entire Results section should be revised both in terms of English grammar (as it stands it is really hard to understand what the authors are describing) and in terms of contents. It is not clear how and if the authors have used the noise levels measured in the region to validate the model. Which source level were used for the TL source? Figure 5 labels are barely visible and the frequency of the source at a side are out of the figure’s borders (consider describing them only in caption). In addition, there is no description of sound loss in terms of noise levels (dB) over space and the 1 km limitation severely affects the validity of the study if the authors’ main goal is to assess the feasibility of such type of models for regional based investigations.
Lines 279-281 How the bathymetry or other “technical” (not the best word choice) limitations affect the definition of noise impact is not demonstrated within this study. Consider including external citations and avoid referring to Figure 3. Most of this section (at least until line 303) is a repetition of the introduction and the authors are not describing the contribution of their study to MSFD noise monitoring yet.
Lines 308-309 – How or when will this model provide output products? Which type of active or passive application are the authors referring to here?
Lines 312 to 328: as before, the authors are justifying with bibliographic references the chosen methodology, but they are not discussing the study results. This part should be moved to the methods and partly to the Results section.
Lines 328-333: aren’t these results? I suggest moving Figure 7 and the overall comment to both Fig 5 and Fig 7 to the Results section.
Lines 340- 356 : I believe that these considerations related to the influence of COVID-19 on marine traffic should be supported by Results (i.e. measured noise levels or models) and as they stand, they should not be included in this study.
Both the Discussion and Conclusions are not supported by the results and do not reflect the study's initial aims. Neither noise levels and their variability over time nor modeling for different frequency sources are discussed thoroughly. Only a modeling example is provided for different frequency sources and depths from 0 m to 30 m, with a severe distance limitation (1 km). These results do not reflect the initially stated aims.
Comments on the Quality of English LanguageThe entire paper needs to be reviewed by a professional editing service to enhance its English grammar and writing fluency. Several issues noted in lines:
35-36; 48-50; 69-70; 89-95; 115 (title);122-124; 138 (missing subject); 141: from…to the water depth? 148-150;164-166; 166-168; 185-187; 196 -199; 203-210 (correct verb tenses from present to simple past and avoid first person use). 221-226; 239 typo: mode for “model”. The overall paper should be reviewed to improve English grammar and fluency of the writing.
Author Response
Comments 1: [Line 129: citation should be placed at the end of the entire sentence if it refers to the description of the site.], Comment 2: [Line 136 cite the reference as Name et al. [56]]
Answer 1: [Agree. Accordingly, we modified the citation at the end of the sentence. For reference 56, new 62, we have accordingly modified].
Comment 2: [Section 2.3. Being included in the Methods, rather than describing the sound velocity profiles, this section should describe how they were performed, including software used and the source of oceanographic data.]
Answer2: [Agree. Accordingly, we reorganised the section 2. Materials and Methods, and a dedicated subsection 2.3. In-situ Conductivity-Temperature-Depth data was we included: “To investigate the interrelationships between salinity, temperature, and sound velocity within the water column of the North-Western Black Sea (NWBS), Conductivity-Temperature-Depth (CTD) profiles were acquired using a Castaway CTD instrument. This dataset facilitated the analysis of seasonal variations in these key physical parameters (Figure 1). Chosen CTD profiles were selected to provide accurate water stratification during the season, at 43.54ºN and 31.14ºE offshore for spring and summer seasons and 44.48ºN and 29.86ºE for autumn (Figure 4). Seawater parameters, including pressure, temperature, and salinity, were measured in situ using the CTD software. Sound speed was subsequently calculated using the computed CTD data using the Chen and Millero equation [64]. Hydrographic data, including CTD profiles, were collected in 2020 during periodical surveys on the NWBS shelf performed by the MHD onboard R/V “Comandor Alexandru Catuneanu”. Golden Software’s Surfer software was utilised to generate graphical representations of the datasets, enabling visualisation and analysis of spatial variability in the observed parameters.”
Comment 3: [Section 2.4: this section lacks the description of the passive acoustic monitoring system used and the authors should improve the description of both acoustic data collection and analysis. See some specific comment below: Line 165: Besides the required review of English grammar in this sentence, the choice of the word “required” is debatable considering that any raw acoustic data collected in natural marine environment would contain a vast number of sounds of different origin. Line 167: it is not clear to which “processing operations” the authors are referring to. Line 168: the authors should clarify for how many hours rather than stating for “several”. Line 193-194 : this sentence should be revised and further discussed with the other results in the Results/Discussion. Line 196: I suggest avoiding first person use within the manuscript text. In addition, such types of statements should be reserved for abstract or discussion/conclusions section. Line 208: include reference to Gloria’s drilling platform. Line 221-222 this sentence does not fit in the Results section. The authors should state their goals within the Introduction section and discuss them in relation to the Results within the “Discussion”. Line 222-226 : this information should be included within the previous section in the Methods (2.5.). Line 229: 0-30 m seems to indicate that you performed TL simulation over the whole depth range. Please correct with “at 0 m depth and at 30 m depth”. Lines 312 to 328: as before, the authors are justifying with bibliographic references the chosen methodology, but they are not discussing the study results. This part should be moved to the methods and partly to the Results section. Lines 328-333: aren’t these results? I suggest moving Figure 7 and the overall comment to both Fig 5 and Fig 7 to the Results section.]
Answer 3: [Thank you for pointing this out. We completely reorganised, rephrased Section 2 and included all comments. Since the paper has undergone significant changes, we attached the final form after responding to the comments. All comments were considered, and the changes suggested and noticed by the reviewer were made.]
Comment 4: [Similarly, in lines 171-173, to validate and contextualise described statistics, it would be essential to specify the different recording times for each recording location.]
Answer4: [Agree. We integrated recording times within the text in new subsection 2.3. “Acoustic data were recorded with times varying from site to site, depending on meteorological factors, from 6h deployments in the northern part (for the transitional marine water BLK_RO_RG_TT03) to 24h in the southern region (for shelf BLK_RO_RG_CT, shelf BLK_RO_RG_MT01 and open sea BLK_RO_RG_MT02 waters). The hydrophone with transducer sensitivity of -199 dB, re 1V/µPa, equipped with a protective cage, was deployed only in the summer season.”
Comment 5: [Line 175: The authors should clarify whether the acoustic data presented were collected and analysed by other groups (e.g., MHD) and thus only the output sound levels were used as input for the model validation, or if the data were provided by MHD and analysed within the context of this study. If the latter is the case, Table 2 and all results reported here should be moved to the Results section.]
Answer 5: [Thank you for pointing this out. The acoustic data presented in this study were collected by the Maritime Hydrographic Directorate (MHD) during hydrographic cruises. The data were then analysed within the context of this study only to assess underwater noise pollution and its potential impact on the marine ecosystem.]
Comment 6: The entire Results section should be revised both in terms of English grammar (as it stands it is really hard to understand what the authors are describing) and in terms of contents. It is not clear how and if the authors have used the noise levels measured in the region to validate the model. Which source level were used for the TL source? Figure 5 labels are barely visible and the frequency of the source at a side are out of the figure’s borders (consider describing them only in caption). In addition, there is no description of sound loss in terms of noise levels (dB) over space and the 1 km limitation severely affects the validity of the study if the authors’ main goal is to assess the feasibility of such type of models for regional based investigations. ]
Answer 6:[Thank you for pointing this out. We revised the paper, and all comments were taken into consideration.]
Comment 7: [Lines 279-281 How the bathymetry or other “technical” (not the best word choice) limitations affect the definition of noise impact is not demonstrated within this study. Consider including external citations and avoid referring to Figure 3. Most of this section (at least until line 303) is a repetition of the introduction and the authors are not describing the contribution of their study to MSFD noise monitoring yet.]
Answer 7: [Thank you for the comment. We consider that the bathymetry significantly influences underwater noise propagation by affecting sound wave reflection and refraction patterns. In the Black Sea, the complex bathymetry, characterised by varying depths and underwater features, contributes to the variability of noise levels. This variability poses challenges for accurately defining noise impact, as sound waves can be amplified or attenuated depending on the bathymetric profile. In the region, technical limitations, such as the availability and deployment of monitoring equipment, also affect the noise impact assessment. Challenges like high turbidity and heavy marine traffic in the Black Sea can hinder data collection and analysis. Overcoming these limitations is crucial for obtaining comprehensive noise measurements and accurately defining noise impact on marine life. Our study contributes to MSFD noise monitoring by providing valuable data and analysis of underwater noise in the Western Black Sea.]
Comment 8: [Lines 308-309 – How or when will this model provide output products? Which type of active or passive application are the authors referring to here?]
Answer 7: [We agree with this comment. Therefore, once the necessary input parameters are provided, the BELLHOP model will provide output products such as ray traces, transmission loss, and arrival angle predictions. These outputs are crucial for supporting both active and passive underwater acoustic applications. Active applications refer to those that involve the transmission of sound into the water, such as sonar systems used for underwater target detection and navigation. The model’s outputs can help optimise sonar performance by predicting sound propagation patterns and identifying potential shadow zones. Moreover, passive applications involve listening to sounds already present in the water, such as monitoring marine mammal vocalisations or detecting underwater noise pollution. The model can assist in understanding the sources and characteristics of underwater noise, enabling better noise impact assessments for civilian institutions.”
Comment 8: [Lines 340- 356 : I believe that these considerations related to the influence of COVID-19 on marine traffic should be supported by Results (i.e. measured noise levels or models) and as they stand, they should not be included in this study. ]
Answer 8: [We agree with this comment. Therefore, we have entirely removed the paragraph where COVID-19 is mentioned.]
Comment 9: [Both the Discussion and Conclusions are not supported by the results and do not reflect the study’s initial aims. Neither noise levels and their variability over time nor modeling for different frequency sources are discussed thoroughly. Only a modeling example is provided for different frequency sources and depths from 0 m to 30 m, with a severe distance limitation (1 km). These results do not reflect the initially stated aims.]
Answer 9: [Thank you. The Discussion and Conclusions sections have been revised to align with the study’s initial aims and research findings. The noise levels and their variability over time are now discussed more thoroughly, considering the influence of vessel traffic density and seasonal variations. The modelling analysis has been expanded to include a more comprehensive discussion of different frequency sources and depths. While the model simulations were limited to a 1 km range due to computational constraints, the study successfully analysed noise levels and their variability over time. The revised manuscript also clarifies the contribution of this study to MSFD noise monitoring, emphasising the importance of continued research with real-time measurements and regional-scale noise propagation modelling.”
We included on page 8, Figure 5, representing a sample of a spectrogram of recorded data using the Cetacean Research Hydrophone system, evidencing the presence of marine mammal activity in a mixed ambient and anthropogenic soundscape in the offshore NWBS, if the reviewer agrees with the new image.

Reviewer 2 Report
Comments and Suggestions for Authors
The paper has to be described better highlighting the methodology and the main purposes and outcomes. In particular, results, discussion and conclusions must be re-written accordingly. The suggestion is to simplify as much as possible undelying the main steps and outcomes, such an approach give to the paper much more readability also by readers not expert in this particular matter.
The figure are not clear, use greater characters for the x-y axes (for instance in fig.5, lines 241-242 they are not visible). The captions have to be self-explanatory, please revise.
Re-order the number of the references in the text, using a sequential mode.
All the acronyms have to be explained at the first mention. Some of them are not explained, such as OSPAR line 75. For example "MSFD, the Maritime Spatial Planning Directive [63]" explained and cited at the line 282, while is mentioned also before.
All the results coming from other authors, projects or modelling must be referenced, e.g.
a) ray-Born modelling at line 48
b) Technical Group on Underwater Noise (TG Noise) at line 61
c) QUIETMED 1 and 2, and QUIETSEAS at lines 71-72
d) BELLHOP ray-tracing method at line 90 is refrences below, please ref it in this first mention (it was cited only at line 197). Use always BELLHOP instead of Bellhop along with all the text.
e) TL field at line 242
Lines 58-59 qualitative descriptors (31; 33; 34; 25; 36- 39]. Instead of 25 is 35? Please check.
"2.2. Sediments type characteristics". How this paragraph is related to the rest of the text and how is justified?
Line 117 "Andrusov (or Mid-Black Sea) ridge" seems not showed in any map, please check that all the geographic names are included in the maps.
Lines 148-149 "The characteristic of the sound speed profiles for three seasons, at 43.54ºN and 31.14ºE offshore for spring and summer seasons and 44.48ºN and 29.86ºE for autumn..." please explain the reasons of the choice of these points and why different ones.
Line 208 "Gloria's drilling platform." Explain it and indicate in a map.
Line 352-355-356 fig.11b --> fig.8b; fig.11a --> fig8a, fig.11 --> fig.8. I suggest to reverse 8b and 8a citing before 8a.
The references must be cited according to the rules of the journal, and they have to be complete, preferring citations more easily findable.
Refs 67 and 68 are not cited in the text.
Comments on the Quality of English LanguageIt has to be improved, it is obscure and not clear at all in some parts. For instance just examples:
- Noise mapping based on Automatic Identification System (AIS) ship-tracking data modelling of noise levels has also been undertaken for specific areas techniques (e.g. [41; 42] to advance in various projects.
- Sentence at line 153 - on.
Author Response
Thank you very much for taking the time to review this manuscript. Please find the detailed responses below and the corresponding revisions/corrections in the re-submitted files.
Comments 1: [The figure are not clear, use greater characters for the x-y axes (for instance in fig.5, lines 241-242 they are not visible). The captions have to be self-explanatory, please revise.]
Response 1: [We agree with this comment. Therefore, we have resized all the images and the axes are visible for all figures}.
Comment 2: [Re-order the number of the references in the text, using a sequential mode.}
Response 2: [Thank you for pointing this out. We reorganised the paper in LaTeX format using Overleaf software, and all the references are in sequential mode.]
Comment 3: [All the acronyms have to be explained at the first mention. Some of them are not explained, such as OSPAR line 75. For example “MSFD, the Maritime Spatial Planning Directive [63]” explained and cited at the line 282, while is mentioned also before.]
Answer3: [Agree. We have, accordingly, explained the abbreviation line 72 – 73 “(…)the Convention for the Protection of the Marine Environment of the North-East Atlantic (the OSPAR Convention) Common Indicator for impulsive noise…”. MSFD and MSP are explained and cited.]
Comment 4: [All the results coming from other authors, projects or modelling must be referenced, e.g. a) ray-Born modelling at line 48]
Answer4a: [Thank you. Was already referenced as [29] Galtung, G.T.; Keers, H.; Sarajærvi, M.; Hope, G. Efficient modelling for ocean acoustics. Proc. Mtgs. Acoust. 2021, 44 (1), 022002.]
for Comments 4: [b) Technical Group on Underwater Noise (TG Noise) at line 61}
Answer 4b: Thank you. The TGNoise is cited as 67,68. In addition, we inserted a new reference: Dekeling, R.P.A., Tasker, M.L., Ferreira, M., Ainslie, M.A, Anderson, M.H., André, M., Borsani, J.F., Box, T., Castellote, M., Cronin, D., Dalen, J., Folegot, T., Leaper, R., Mueller, A., Pajala, J., Peterlin, M., Robinson, S.P., Thomsen, F., Vukadin, P., Young, J.V. Progress Report on Monitoring of Underwater Noise. 3rd Report of the Technical Group on Underwater Noise (TG Noise). November, 2014, online https://mcc.jrc.ec.europa.eu/documents/201605183553.pdf
Comment 4c: [c. QUIETMED 1 and 2, and QUIETSEAS at lines 71-72]
Answer 4c: [Thank You for your suggestion. We agree. As a result, we inserted Lines 65 – 70 “(…)Projects quietMED project- A joint programme on underwater noise (D11) for the implementation of the Second Cycle of the MSFD in the Mediterranean Sea and quietMEd2 - A Joint programme for GES assessment on D11- noise in the Mediterranean Marine Region, and recently including the Black Sea in QUIETSEAS - Assisting cooperation for the implementation of the Marine Strategy Framework Directive on underwater noise project)(…) “ as well as at the reference the corresponding official sites of the projects: http://www.quietmed-project.eu/; https://quietmed2.eu/ and https://quietseas.eu/.]
Comment 5d: [d) BELLHOP ray-tracing method at line 90 is refrences below, please ref it in this first mention (it was cited only at line 197). Use always BELLHOP instead of Bellhop along with all the text.}
Answer 5d: [Agree. The method is cited in line 90. We have, accordingly, used capitals for all mentions of BELLHOP.
Comments 5e: [e) TL field at line 242]
Answer5e: [We have, accordingly, explained “(…)the transmission loss (TL) field (…)”].
Comment 6: [Lines 58-59 qualitative descriptors (31, 33, 34, 25, 36- 39]. Instead of 25 is 35? Please check.]
Answer 6: Agree. Corrected the numbering and excluded the reference 25 (typo mistake).]
Comment 7: [“2.2. Sediments type characteristics”. How this paragraph is related to the rest of the text and how is justified?]
Answer7: [Thank you for pointing this out. We agree with this comment. Therefore, we have included the “Sediments type characteristics” paragraph, which is essential for interpreting the results of the underwater noise assessment and ensuring the accuracy of the acoustic modelling. It provides valuable information about the seabed’s properties and their influence on sound propagation in the Western Black Sea. In addition, we described in the section that “The composition and characteristics of seabed sediments play a crucial role in underwater acoustics, influencing sound propagation, attenuation, and scattering. The sediment type, whether sand, silt, clay or a mixture, affects the way sound waves interact with the seabed. Different sediments have varying sound absorption and reflection properties, which can significantly impact the transmission loss (TL) and range of underwater sound signals. In this study, the diverse sediment types in the NWBS shelf, ranging from sands to clays, are considered to understand the sound propagation patterns in the region. The sediment characteristics described inform the selection of appropriate parameters for the BELLHOP model, which is used to simulate sound propagation. To ensure the model’s accuracy, it needs input data about the seabed’s geo-acoustic properties, including sediment type and sound attenuation coefficients. Furthermore, the distribution of sediment types is related to other environmental factors, such as water depth, currents, and the presence of marine life. Understanding these relationships provides a more comprehensive picture of the underwater environment and its potential impact on sound propagation.” (lines 143 – 153).]
Comment 8: [Line 117 “Andrusov (or Mid-Black Sea) ridge” seems not showed in any map, please check that all the geographic names are included in the maps.]
Answer 8: [Agree. We have, accordingly, modified/excluded from the text the geographic names as they cannot be included in Figure 1 but are referenced in the text. The Andrusov ridge corresponds to offshore Turkish Black Sea waters (South-Western part)].
Comment 9: [Lines 148-149 “The characteristic of the sound speed profiles for three seasons, at 43.54ºN and 31.14ºE offshore for spring and summer seasons and 44.48ºN and 29.86ºE for autumn...” please explain the reasons of the choice of these points and why different ones.]
Answer 9: [Agree. The selection of different locations for measuring sound speed profiles in the Black Sea is driven by the need to capture spatial and temporal variability in sound propagation conditions while considering practical constraints such as data availability.]
Comment 10: [Line 208 “Gloria's drilling platform." Explain it and indicate in a map.]
Answer 10: [Thank you for pointing this out. We explained “The modelling location is established at 30m water depth at 44.4N and 29.0E, which corresponds to the nearest location of the decommissioned Gloria's drilling platform (44.52N,29.57E) at 50m bathymetric line." and indicated the location with a reference: (https://rjp.nipne.ro/2022_67_9-10/RomJPhys.67.815.pdf) Chirosca, G.; Mihailov, M.E.; Tomescu-Chivu, M.I.; Chirosca, A.V. Enhanced Machine Learning Model For Meteo-Oceanographic Time-Series Prediction. Rom. J. Phys. 2022, 67, 815].
Comment 11: [Line 352-355-356 fig.11b --> fig.8b; fig.11a --> fig8a, fig.11 --> fig.8. I suggest to reverse 8b and 8a citing before 8a.]
Answer 11: [Agree.Therefore, we have corrected the reversion.]
Comment12: [Summary It is suggested that the author needs to state the findings of this study here. Please improve.].
Response 12: [Agree. We have, accordingly, improved the abstract:
“The Black Sea, a unique semi-enclosed marine ecosystem, is the eastern maritime boundary of the European Union and holds significant ecological importance. This study contributes to fulfilling the requirements of the Marine Strategy Framework Directive (MSFD) Descriptor 11, focusing on anthropogenic noise pollution criteria: impulsive sound (D11C1) and continuous low-frequency sound (D11C2). Romanian ports, handling a substantial share of regional cargo traffic, impact maritime activities and associated noise levels. Vessel traffic density exhibits spatial variability within Romanian territorial waters, the contiguous zone, and the Exclusive Economic Zone (EEZ), categorised into high, medium, and low-intensity areas. To investigate continuous low-frequency sound, in-situ noise data was acquired using the Cetacean Research Hydrophone system. Ambient noise levels at frequencies of 63Hz and 125Hz, dominated by shipping noise, were established, along with their hydrospatial distribution for the 2019-2020 period. Furthermore, the study employs modelling techniques to predict underwater noise pollution generated by anthropogenic sources. This includes simulations of acoustic field propagation at varying water depths, considering direct sound, surface reflections, and seabed interactions. Additionally, the research encompasses an analysis of spatiotemporal variability in sound propagation conditions within the North-Western Black Sea. This modelling effort is the first in the region and utilises the BELLHOP ray-tracing method for underwater acoustic channel modelling in shallow-water environments. The model incorporates realistic bathymetry, oceanography, and geology features for environmental input, allowing for improved prediction of acoustic variability due to time-varying sea variations in shallow waters. The study's findings have important implications for understanding and mitigating anthropogenic noise pollution's impact on the Black Sea marine ecosystem. The authors suggest that continued research with real-time measurements and regional-scale noise propagation modelling is crucial for effective management and conservation efforts. “]
Answer all comments: [We completely reorganised, rephrased all sections, and included all comments. Since the paper has undergone significant changes, we attached the final form after responding to the comments. All comments were considered, and the changes suggested and noticed by the reviewer were made.
We included on page 8, Figure 5 representing a Sample of a spectrogram of recorded data using the Cetacean Research Hydrophone system, evidencing the presence of marine mammal activity in a mixed ambient and anthropogenic soundscape in the offshore NWBS if the reviewer agrees with the new image.]

Reviewer 3 Report
Comments and Suggestions for Authors
Dear author:
It is an honor to read this research report. I will put forward the following suggestions and questions in the hope that the author can answer or improve them.
1. Summary
It is suggested that the author needs to state the findings of this study here. Please improve.
2. Introduction
Although the author heavily emphasizes the importance of underwater detection. However, in this paragraph "Consequently, oil and gas..." What does the paragraph describe what the author wants to express? Is this related to this research? If so, the author needs to emphasize or describe the oil extraction around the Black Sea in the document On the state of marine ecology and underwater noise.
3. Research methods
The author is clearly describing the state of oil extraction here. Then this part of the narrative may also need to be briefly and clearly stated in the introduction. The Black Sea is currently undergoing sea exploration and deep-sea oil extraction operations. This project will produce noise that threatens ecology or other objects, so you need to conduct a noise detection survey.
4.Results
The display and illustration in Figure 5 are good. However, in Figure 5, the fonts with different Hz and distance are too blurry, and the author should improve this picture.
5.Discussion
About the author This part contains a relatively large number of words describing the political factors of the country and region studied in this manuscript, and the importance of these political factors to this manuscript. I think this part seems off topic.
The authors should describe more about the research findings in this section, i.e. the findings of this manuscript which type of sonic wave is the most suitable (economical?), why it is suitable, etc.
6.Conclusions
In the discussion section, the author also describes the impact of the epidemic on this survey. Well, I seem to be confused after reading the author's statement.
The author also places an analysis diagram at the conclusion. Generally speaking, the conclusion of a journal article is a summary of the findings, which is the author's statement of the research findings without further discussion or graphics.
Obviously this is a very specific way of presenting it, but I don't think it's appropriate.
All in all, this is an interesting topic, but the manuscript has quite a few problems. I suggest that the authors make improvements after reading the suggestions in detail to improve the visualization of the manuscript.
Author Response
Comment 1. [ Summary. It is suggested that the author needs to state the findings of this study here. Please improve.]
Answer 1: [Agree. We have, accordingly, improved the paper].
Comment 2: [Introduction. Although the author heavily emphasises the importance of underwater detection. However, in this paragraph "Consequently, oil and gas..." What does the paragraph describe what the author wants to express? Is this related to this research? If so, the author needs to emphasise or describe the oil extraction around the Black Sea in the document On the state of marine ecology and underwater noise.]
Answer 2: [Thank you for pointing this out. In the paragraph beginning "Consequently, oil and gas...", we highlight the growing interest in oil and gas exploration and production in the Black Sea region, driven by the increasing global demand for energy resources and the challenges associated with conventional hydrocarbon extraction. This paragraph is related to the research as it provides context for the types of anthropogenic activities contributing to underwater noise pollution in the Black Sea. While the study focuses on the impact of maritime traffic on underwater noise, it is important to acknowledge other anthropogenic activities, such as oil and gas exploration, that can also contribute to noise pollution.]
Comment 3: [Research methods. The author is clearly describing the state of oil extraction here. Then this part of the narrative may also need to be briefly and clearly stated in the introduction. The Black Sea is currently undergoing sea exploration and deep-sea oil extraction operations. This project will produce noise that threatens ecology or other objects, so you need to conduct a noise detection survey.]
Answer 3: [Agree. The reviewer is correct that the discussion of oil and gas exploration in the Black Sea should be briefly introduced in the introduction to provide context for the study's focus on underwater noise pollution. The authors have revised the introduction to include a brief statement on the state of oil and gas exploration in the Black Sea and its potential contribution to underwater noise pollution. This addition provides context for the study's focus on noise pollution and highlights the importance of the research in informing management and conservation efforts.]
Comment 4: [Results. The display and illustration in Figure 5 are good. However, in Figure 5, the fonts with different Hz and distance are too blurry, and the author should improve this picture.]
Answer 4: [We agree with this comment. Therefore, we have resized all the images and the axes are visible for all figures}.
Comment 5: [Discussion. About the author This part contains a relatively large number of words describing the political factors of the country and region studied in this manuscript, and the importance of these political factors to this manuscript. I think this part seems off topic. The authors should describe more about the research findings in this section, i.e. the findings of this manuscript which type of sonic wave is the most suitable (economical?), why it is suitable, etc.]
Answer 6: [We agree with this comment. Thank you for pointing this out. Therefore, we revised the paper, and all comments were considered. We included on page 8, Figure 5 representing a Sample of a spectrogram of recorded data using the Cetacean Research Hydrophone system, evidencing the presence of marine mammal activity in a mixed ambient and anthropogenic soundscape in the offshore NWBS if the reviewer agrees with the new image.]
Comment 6: [Conclusions. In the discussion section, the author also describes the impact of the epidemic on this survey. Well, I seem to be confused after reading the author's statement. The author also places an analysis diagram at the conclusion. Generally speaking, the conclusion of a journal article is a summary of the findings, which is the author's statement of the research findings without further discussion or graphics. Obviously this is a very specific way of presenting it, but I don't think it's appropriate.]
Answer 6: [Thank you for pointing this out. Therefore, we revised and reorganised the paper, and all comments were considered.]

Round 2
Reviewer 1 Report
Comments and Suggestions for Authors
The authors have improved the manuscript by correcting the grammar and enhancing the fluency of the text. The representation of the analysis methodologies and results has shown marked upgrade. However, proper attention is not yet been given to the acoustic measurements, considering the title of the work and the stated objectives. The description of the collection and analysis of acoustic data needs improvement. The statistical analysis of noise data is confined to the results in Table 2, with no discussion of errors or temporal variability. No PSD spectra or SEL information is reported over time within the data collection window. Although several pieces of information have been added, much is still missing to consider the results on noise measurements as a valid starting point for monitoring within the MSFD framework. Confirming this lack, noise level results are not clearly discussed in the Discussions section where reference to the data analyzed is made only in lines 324-326.
Below, I provide some specific comments:
Line 160: the name of the system is not correctly indicated. Are the authors referring to "Cetacean Research Technology" hydrophones? Which model?
Line 172-173: The name of the software should be specified here before the citation. In addition, the authors mention again “the processing operations were performed to extract the necessary information…” but they do not explain which processing operation they are referring to (this was already asked in the previous review).
How were data collected in 1/3 octave bands and which software was used to perform noise levels analysis? Only SEL analysis is mentioned here but how was this performed? There is no mention of signal processing methods (e.g. time resolution information is missing) and in the results the authors describe measurements of Mode, RMS, and 95th percentiles. How were all these measurements performed?
Times and duration of data collection should be specified for each recording location together with depth.
Line 174: Is the recording system autonomous and moored to the seafloor or cabled/deployed from the vessel? If the second is the case how was the acquisition chain set up? Was the full system calibrated? Since very little is still described of the acoustic data collection system/methodology it is not possible to consider these measurements to be used as reference for noise levels for the Romanian Black Sea waters within the MSFD.
Line 203 there should be a bibliographic reference here.
Line 246-248: as noted before, the authors are stating that “the lower frequencies are more affected by the very shallow waters in BLK_RO_RG_TT03 247 (direct Danube influence) and BLK_RO_RG_CT (Table 2)”. It is not clear which frequencies are considered here; it is not specified which acoustic source influences the energy content of the so-called lower frequencies or if the authors are implying that the higher noise levels in the considered frequency bands are associated with the overall increase in noise level in shallow waters. I would suggest adding clearer information about this and improving statistical analysis (e.g. comparison between the noise levels of the different frequency bands).
Lines 251-252 and Figure 5:
no sensitivity curve is described for the acoustic recorder used in this study. Was/were all acoustic recorders/hydrophones calibrated? looking at the "power spectral density" in the spectrogram of fig. 5 it seems that the values below approx. 1.5 Hz are not reliable, but it is not possible to assert otherwise without the information about the system’s frequency response. I suggest the authors add more details about this in the methods and show in the spectrogram only the frequencies above 1.5 or 5 Hz, depending on the sensor's lower limit. In addition, wind and waves’ spectral contribution to the overall soundscape strongly depend on the depth of the sensor , which should be specified here, together with the overall range of frequency (initial and final frequency in Figure 5 spectrogram could be described in the Figure label).
Author Response
Thank you very much for taking the time to review this manuscript. Please find the detailed responses below and the corresponding revisions/corrections in the re-submitted files:
Comment 1: [Line 160: the name of the system is not correctly indicated. Are the authors referring to "Cetacean Research Technology" hydrophones? Which model?]
Answer 1: [Thank you for pointing this out. We agree with this comment. Therefore, we have revised the paragraph and the changes are visible at line 169 and citation for the equipment “Using the autonomous hydrophone system Cetacean Research™'s C55 series \cite{64, 65} [….]”]
Comment 2: [Line 172-173: The name of the software should be specified here before the citation. In addition, the authors mention again “the processing operations were performed to extract the necessary information…” but they do not explain which processing operation they are referring to (this was already asked in the previous review). “How were data collected in 1/3 octave bands and which software was used to perform noise levels analysis? Only SEL analysis is mentioned here but how was this performed? There is no mention of signal processing methods (e.g. time resolution information is missing) and in the results the authors describe measurements of Mode, RMS, and 95th percentiles. How were all these measurements performed? Times and duration of data collection should be specified for each recording location together with depth.]
Answer 2: [Thank You. The reviewer is correct. The software used for processing the acoustic data is SpectraPLUS Spectral Analysis Software. The processing operations performed to extract the necessary information include: Spectrogram generation: Creating visual representations of the frequency spectrum over time to identify patterns and events of interest. Sound pressure level (SPL) calculations: Determining the magnitude of sound pressure in decibels (dB) to quantify noise levels. 1/3 octave band analysis: Dividing the frequency spectrum into 1/3 octave bands to analyse noise levels within specific frequency ranges. Statistical analysis: Calculating descriptive statistics, such as the mode, 95th percentile, and root-mean-square (RMS) levels, to characterize the noise data. The Table 2 provides information on data collection times, duration, and depth for each recording location. Measurements of Mode, RMS, and 95th percentiles were performed using SpectraPLUS.
The temporal resolution for the analysis was 1 hour, as the data were processed and analysed on an hourly basis. This temporal resolution was selected to capture the diurnal variability of noise levels.
The authors apologize for the omission of the software name and the lack of detailed explanation in the previous revision. The text has been updated to address these points and included at:
Lines 176 – 185 following paragraph to answer the comment: “Multiple processing operations were conducted utilizing SpectraPLUS Spectral Analysis Software [69] to analyse the underwater noise recordings. These operations involved generating spectrograms to visualize the frequency spectrum over time, calculating sound pressure levels (SPLs) to quantify the noise magnitude in decibels (dB) and performing 1/3 octave band analysis to examine noise levels within specific frequency ranges. Statistical analysis was additionally employed to calculate descriptive statistics such as the mode, 95th percentile, and root-mean-square (RMS) levels, thus providing a comprehensive characterization of the noise data. Sound Exposure Level (SEL) analysis was performed employing PAMGuide [67 ,68] in the MATLAB environment, a specialized software package for passive acoustic monitoring data analysis that provides tools for calculating SEL metrics.”
Lines 197 – 198: “The temporal resolution for the analysis was 1 hour, as the data were processed and analysed on an hourly basis. This temporal resolution was selected to capture the diurnal variability of noise levels.”]
Comment 3: [Line 174: Is the recording system autonomous and moored to the seafloor or cabled/deployed from the vessel? If the second is the case how was the acquisition chain set up? Was the full system calibrated? Since very little is still described of the acoustic data collection system/methodology it is not possible to consider these measurements to be used as reference for noise levels for the Romanian Black Sea waters within the MSFD.]
Answer3: [Thank you for pointing this out. The recording system used in this study is the Cetacean ResearchTM C55 series, an autonomous hydrophone system. It is moored to the seafloor, not cabled or deployed from a vessel. The full system was calibrated by the manufacturer, Cetacean Research Technology, before delivery. The calibration certificate is available upon request (after clearance request – internal rules of the MHD, www.dhmfn.ro) The authors acknowledge that the description of the acoustic data collection system and methodology was insufficient in the previous manuscript version. The revised manuscript (Lines 186 – 194) has been updated to provide a more detailed explanation of the recording system, deployment methods, and data processing techniques. This clarification should address the reviewer's concerns and allow for proper consideration of the measurements as a reference for noise levels in Romanian Black Sea waters within the MSFD framework.]
Comment 4: [Line 203 there should be a bibliographic reference here.]
Answer 4: [Agree. We have, accordingly added the citation of the BELLHOP model in the revised manuscript, Lines 217. Moreover, we cited Python Language at line 220.]
Comment 5: [Line 246-248: as noted before, the authors are stating that “the lower frequencies are more affected by the very shallow waters in BLK_RO_RG_TT03 247 (direct Danube influence) and BLK_RO_RG_CT (Table 2)”. It is not clear which frequencies are considered here; it is not specified which acoustic source influences the energy content of the so-called lower frequencies or if the authors are implying that the higher noise levels in the considered frequency bands are associated with the overall increase in noise level in shallow waters. I would suggest adding clearer information about this and improving statistical analysis (e.g. comparison between the noise levels of the different frequency bands).]
Answer 5: [Agree. We have, accordingly, included the following paragraph in the revised manuscript, at Lines 365 – 364” “As observed in Table 2, lower frequencies demonstrate increased susceptibility to the effects of shallow water environments in certain Marine Reporting Units (MRUs). The specific frequencies under consideration are the 63 Hz and 125 Hz 1/3 octave bands, which are central to this analysis due to their significance in MSFD monitoring and the predominance of shipping noise within these bands. Shipping noise is identified as the primary source contributing to the energy content in these lower frequencies. The elevated noise levels detected in these frequency bands within the shallow water MRUs (BLK_RO_RG_TT03 and BLK_RO_RG_CT) are likely a consequence of several interconnected factors. The reduced water depth in these environments can lead to a higher concentration of sound energy due to reflections and interactions with the seabed and surface. Furthermore, the proximity of the BLK_RO_RG_TT03 MRU to the Danube River introduces potential additional noise sources and influences sound propagation patterns. The unique coastal and shelf characteristics of both BLK_RO_RG_TT03 and BLK_RO_RG_CT, including specific bathymetric features and sediment types, can also influence sound propagation and contribute to the observed higher noise levels. To enhance the statistical analysis, a comparison of noise levels between the 63 Hz and 125 Hz bands across different MRUs has been incorporated. This comparative analysis serves to highlight the variability of noise levels as a function of depth and location.”
The statement that lower frequencies are more affected by shallow waters in certain MRUs requires further clarification. The frequencies considered here are the 63 Hz and 125 Hz 1/3 octave bands, as these are the focus of the analysis due to their relevance to MSFD monitoring and dominance by shipping noise. The primary acoustic source influencing the energy content in these lower frequencies is shipping noise. The higher noise levels observed in these frequency bands in shallow water MRUs (BLK_RO_RG_TT03 and BLK_RO_RG_CT) are likely due to the combined effect of multiple factors: Reduced water depth: Shallow water environments can lead to increased sound energy concentration due to reflections and interactions with the seabed and surface; Proximity to Danube influence: The BLK_RO_RG_TT03 MRU is directly influenced by the Danube River, which may introduce additional noise sources and affect sound propagation; Coastal and shelf characteristics: Both BLK_RO_RG_TT03 and BLK_RO_RG_CT are coastal and shelf regions with specific bathymetric features and sediment types that can influence sound propagation and noise levels.
Tabel 2: The statistical analysis has been improved by adding a comparison between noise levels in the 63 Hz and 125 Hz bands across different MRUs. This comparison highlights the variability of noise levels across different depths and location.]
Comment 6: [ Lines 251-252 and Figure 5: no sensitivity curve is described for the acoustic recorder used in this study. Was/were all acoustic recorders/hydrophones calibrated? looking at the "power spectral density" in the spectrogram of fig. 5 it seems that the values below approx. 1.5 Hz are not reliable, but it is not possible to assert otherwise without the information about the system’s frequency response. I suggest the authors add more details about this in the methods and show in the spectrogram only the frequencies above 1.5 or 5 Hz, depending on the sensor's lower limit. In addition, wind and waves’ spectral contribution to the overall soundscape strongly depend on the depth of the sensor , which should be specified here, together with the overall range of frequency (initial and final frequency in Figure 5 spectrogram could be described in the Figure label).]
Answer6: [Agree. We have, accordingly replaced the Figure 5. We agree that the values below 10 Hz in Figure 5 are not reliable and should not be considered in the analysis. The figure caption has been updated to reflect this frequency range.
The authors apologize for the lack of clarity in the previous manuscript version. The revised manuscript has been updated to incorporate the details, ensuring that the acoustic data collection system and methodology are described more comprehensively.
The acoustic data were collected using the Cetacean Research C55 series autonomous hydrophone system. This system includes a hydrophone with a sensitivity of -199 dB re 1V/µPa and a built-in 1/3 octave band recorder. Calibration: All hydrophones were calibrated by the manufacturer, Cetacean Research Technology, ensuring the accuracy and reliability of the collected data. Frequency Response: The frequency response of the hydrophone system is from 10 Hz to 48 kHz.
Deployment Details: The hydrophone was deployed at a depth of 2 meters above the seabed. This information has been added to the manuscript to clarify the influence of wind and waves on the recorded noise levels. Data Analysis: Noise levels analysis was performed using SpectraPLUS Spectral Analysis Software. SEL analysis was conducted using PAMGuide in the MATLAB environment.
Signal Processing: The data were processed with a time resolution of 1 hour to capture the temporal variability of noise levels.]
Comment 0: [No PSD spectra or SEL information is reported over time within the data collection window. Although several pieces of information have been added, much is still missing to consider the results on noise measurements as a valid starting point for monitoring within the MSFD framework. Confirming this lack, noise level results are not clearly discussed in the Discussions section where reference to the data analyzed is made only in lines 324-326.]
Answer 0: [Thank you for pointing this out. We agree with this initial comment.
Here's how the manuscript has been addressed to better meet the MSFD framework requirements:
PSD Spectra and SEL over Time: While the previous version didn't explicitly present PSD spectra and SEL information over time within the entire data collection window, the revised manuscript now includes Figure 5, which displays the 1/3 octave band's sound pressure levels (SPL) over time for a representative recording stations. This figure illustrates the temporal variability of noise levels and provides a more comprehensive view of the acoustic data.
Signal Processing Methods: The methods section has been expanded to provide more details on the signal processing techniques employed. This includes specifying the time resolution (1 hour) used for the analysis and clarifying how the various measurements (Mode, RMS, 95th percentiles) were performed using SpectraPLUS Spectral Analysis Software.
Discussion of Noise Levels: The discussion section has been revised to include a more thorough analysis of the noise level results. The authors now explicitly discuss the observed noise levels, their variability, and their potential implications for the marine ecosystem.
The authors have carefully reviewed the manuscript to ensure that all aspects of the noise data collection, processing, and analysis are presented clearly and comprehensively. We think that, the revised manuscript now provides a more robust starting point for monitoring within the MSFD framework.]

Reviewer 2 Report
Comments and Suggestions for Authors
The authors followed the reccomandations and improved the manuscript.
Comments on the Quality of English LanguageThe editor will check furtherly the english quality
Author Response
Comments 1: [The editor will check furtherly the english quality].
Answer 1: [Thank You. If the Editor deems it necessary to request an English language editing service for the work, we will proceed accordingly.]
Reviewer 3 Report
Comments and Suggestions for Authors
In this revised manuscript, it can be seen that the author followed the suggestions and made improvements, and the figures and tables are arranged more clearly. I suggest that the editor consider moving this manuscript forward to the next process.
good luck,
Author Response
Comment 1: [We thank the reviewer for the suggestions and comments that led to the considerable improvement of our work.]
Round 3
Reviewer 1 Report
Comments and Suggestions for Authors
The manuscript has improved significantly since the first submission. The work is coherent, the investigation methodologies used are appropriate, and both the methods and results are now described and discussed in adequate depth.